Porites superfusa mortality and recovery from a bleaching event at Palmyra Atoll, USA

Furby Kathryn Anne kfurby@ucsd.edu
Smith Jennifer Ellen
Sandin Stuart Adrian
Scripps Institution of Oceanography, University of California San Diego , San Diego , United States
Reimer James
Electronic publication date: 2017 May 2
Publication date: 2017
Volume: 5
Electronic Location ID: e3204
Received 2016 Aug 1; Accepted 2017 Mar 20
Copyright: ©2017 Furby et al.
Copyright year: 2017
Copyright holder: Furby et al.
License: This is an open access article distributed under the terms of the Creative Commons Attribution License, which permits unrestricted use, distribution, reproduction and adaptation in any medium and for any purpose provided that it is properly attributed. For attribution, the original author(s), title, publication source (PeerJ) and either DOI or URL of the article must be cited.
License URL: https://creativecommons.org/licenses/by/4.0/

Keywords: Climate change, Coral bleaching, Population dynamics, ENSO, Coral regrowth, Coral recovery

Funding: Gordon and Betty Moore Foundation National Science Foundation Graduate Research Fellowship 2011116612 Scripps Family Foundation United States Fish and Wildlife Nature Conservancy Funding and logistical support for this research was provided by the Gordon and Betty Moore Foundation, the National Science Foundation Graduate Research Fellowship (2011116612), the Scripps Family Foundation, United States Fish and Wildlife, and the Nature Conservancy. The funders had no role in study design, data collection and analysis, decision to publish, or preparation of the manuscript.

==============================
Background

The demography of a coral colony is not a binary trajectory of life and death. Based on the flexibility afforded by colonial organization, most reef-building corals employ a variety of dynamic survival strategies, including growth and shrinkage. The demographic flexibility affects coral size, shape and reproductive output, among other factors. It is thus critical to quantify the relative importance of key dynamics of recruitment, mortality, growth and shrinkage in changing the overall cover of coral on a reef.

Methods

Using fixed photographic quadrats, we tracked the patterns of change in the cover of one common central Pacific coral, Porites superfusa, before and after the 2009 ENSO event.

Results

Coral colonies suffered both whole and partial colony mortality, although larger colonies were more likely to survive. In subsequent years, recruitment of new colonies and regrowth of surviving colonies both contributed to the modest recovery of P. superfusa.

Discussion

This study is unique in its quantitative comparisons of coral recruitment versus regrowth during periods of areal expansion. Our data suggest that recovery is not limited simply to the long pathway of settlement, recruitment and early growth of new colonies but is accelerated by means of regrowth of already established colonies having suffered partial mortality.

Introduction

Understanding patterns of mortality and recovery among reef-building corals is foundational to the development of accurate predictions of coral population trajectories (Baird & Marshall, 2002). Recruitment has been recognized as important for long-term recovery after major and frequent disturbances, especially for coral populations that have suffered widespread, whole-colony mortality (Dollar & Tribble, 1993). However, many disturbance events are less extreme, leading to a combination of partial and complete mortality. We define partial mortality of colonial corals as the response to a stress event that leads to tissue loss, but with the survival of some colony area. For colonies suffering from partial mortality, regeneration of tissue following stress is a critical mechanism in assuring individual colony survival (Chadwick & Loya, 1990). As such regrowth of corals has been suggested as an important mechanism in coral population recovery (Diaz-Pulido et al., 2009; Gilmour et al., 2013). In this context, analyzing recovery characteristics can help project population resilience.

Given the colonial nature of most reef-building corals, the definition of an ‘individual’ presents real challenges (Hughes, Ayre & Connell, 1992). Coral ecological literature often refers to an individual as a colony. Interactions and population dynamics are sometimes best described at the colony scale, including patterns of size-specific mortality, competition for space on the benthos, and vulnerability to storm damage (e.g., Hughes & Jackson, 1980; McCook, Jompa & DiazPulido, 2001; Baird & Marshall, 2002; Madin et al., 2014). In contrast the physiological definition of an individual is the polyp (Harper, 1985). In particular both asexual (fission/ budding) and sexual (spawning/ brooding) modes of reproduction occur at the scale of the polyp. When tracking demographic responses to environmental change or disturbance, it is critical to account for dynamics occurring at both the scale of the colony and the scale of the polyp.

Corals are capable of four known mechanisms of loss and recovery: complete-colony mortality, partial colony mortality (tissue loss), recruitment, and partial-colony growth (sometimes referred to as regrowth) (Fig. 1). A few recent studies have documented the relative importance of recruitment and regrowth as mechanisms of coral population recovery following a disturbance (Diaz-Pulido et al., 2009; Gilmour et al., 2013; Roff et al., 2014); however, the colonial mechanisms of coral recovery have not been regularly quantified.

Figure 1 Conceptual diagram of coral demographics contributing to total coral cover for a population.

Blue arrows denote contribution to increasing coral cover (e.g., True recruits, ‘Resurrected’ recruits, and Growth). Orange arrows denote contribution to decreasing coral cover (e.g., Partial mortality and Complete mortality). ‘Resurrected recruits’ is a new term indicating an operationally defined form of regrowth (see Methods for detailed description). The thickness of arrows indicates the mechanism’s relative contribution, as found in this study.

Due to its remote location and lack of local anthropogenic impacts, Palmyra Atoll is an ideal location for studying demographic rates of corals in response to global change (Knowlton & Jackson, 2008; Sandin et al., 2008; Williams et al., 2011). Palmyra is a US National Wildlife Refuge and is part of the Pacific Remote Island Areas National Marine Monuments and is protected from fishing and other impacts. In 2009, Palmyra Atoll experienced a mild bleaching event associated with an El Nino Southern Oscillation (ENSO) event (Williams et al., 2010). During the 2009 ENSO event, sea surface temperatures reached 1.5 °C above the maximum long-term monthly temperatures, and the anomaly continued for four months (Williams et al., 2014). October 2009 through March 2010 had over four DHW (degree heating weeks) with November through March over 8 DHW (NOAA Coral Reef Watch).

This study examines patterns of change in an encrusting coral, Porites superfusa, on Palmyra Atoll during and after the thermal stress event. Specifically, our objectives were to address two complementary questions: (1) Does P. superfusa cover change through time from 2009 to 2012 on Palmyra Atoll; (2) What are the relative rates of colony survivorship (recruitment/ mortality) versus colony growth (growth/shrinkage) during and after a bleaching event?

Materials & Methods

Study species

Porites superfusa is a small coral that is relatively ubiquitous on Palmyra Atoll, reaching almost 400 cm2 in area with an overall mean colony size of 9.9 cm2 (median = 4.4 cm2). Our targeted design involved this species because it is one of the most numerous corals in the area, and its compact morphology allows study of multiple individuals within square meter photoquadrats (see below). In 2009, the P. superfusa population showed extensive evidence of bleaching with a notable reduction in cover in the permanent photoquadrats.

Surveys

Four forereef sites on Palmyra Atoll were selected, two sites on the north and two sites on the south shore, each approximately 2 km apart (Fig. 2). Sites were chosen to be representative of Palmyra’s forereef habitats. These sites all had high initial densities (C2009) of Porites superfusa colonies, and as they are evenly spaced across the island they should capture within-island differences in the forereef habitat. Sites were surveyed four times at approximately annual intervals (September 2009, July 2010, September 2011, and September 2012).

Figure 2 Study sites (FR3, FR5, FR7, FR9) around Palmyra Atoll.

Palmyra is located in the remote central Pacific. The black area shows the atoll land area and the gray and white areas denote different depth strata of reef and open ocean.

Figure 3 Representative permanent photoquadrat sequence from one site (FR9).

(A) September 2009 (C2009) with living Porites superfusa outlined in black. (B) July 2010 (C2010) with previous 2009 colony area outlined in black, and living coral highlighted in red. (C) September 2011 (C2011) with previous 2009 colony area outlined in black, 2010 colony area outlined in red, and living coral highlighted in yellow. Colonies that show examples of different fates are labeled with numbers: (1) complete mortality, (2) partial mortality, (3) true recruitment, (4) growth, and (5) resurrected recruits. Quadrats are 0.54 m2.

At each site, one 50 m transect was permanently marked, parallel to shore and at a depth of 10 m. In 2009, ten permanent photoquadrats were established at each site, positioned every 5 m along the transects. The corners of plots were marked with stainless steel eyebolts held in place with marine epoxy (US Fish and Wildlife Special Use Permit 12533-16006). Each photoquadrat was imaged with a Canon G12 camera attached to a PVC frame (0.54 m2) by SCUBA divers (Sandin et al., 2008). These marked plots were revisited and re-surveyed, enabling the tracking of individual colony fates through time. In sum, 40 0.54 m2 plots were surveyed annually for a total of four time points each.

Image analyses

Within each photograph, colonies of P. superfusa ranged in area from 0.2 cm2 to 440 cm2. Due to the limitations of the photographic resolution, the tracking of the smallest colonies was not possible and thus we focused our study on colonies exceeding 1 cm2. Photographs were analyzed using ImageJ to calculate the size (estimated as 2-dimensional area when viewed from the top down) and survivorship of each P. superfusa colony within the images (Fig. 3, Abramoff, Magalhaes & Ram, 2004). Each colony was tagged digitally and tracked through time (similar to methods of Hughes & Jackson, 1985). Fates of the colonies were placed into one of five categories—complete mortality, partial mortality, true recruits, growth, and ‘resurrected’ recruits. Complete mortality was defined as the death of the entire visible coral, with its previous location overgrown by other organisms (Fig. 3B-1). Partial mortality (i.e., injury, shrinkage) was recorded when colonies lost tissue (Fig. 3B-2). True recruitment (or settlement) indicated a new coral recruit had claimed substrate in an area previously without P. superfusa (Fig. 3B-3). Growth occurred when a previously present colony created additional tissue (Fig. 3B-4). ‘Resurrected’ recruits were defined as apparent recruits that appear in a location where a colony had been recorded as suffering mortality in previous time points (Figs. 3C-5 and 1). Because of the limitations of the photographic census, it was impossible to determine whether such recruits were new recruits (new colony settling at the same location) or regrowth from microscopic areas of cryptic remnant tissue. Individual colony cover and total P. superfusa cover were calculated per quadrat. Data are reported in two formats: colony-specific fates and total live P. superfusa cover summed within quadrats tracked through time.

Data analyses

Analyses assessed patterns of variation in starting live P. superfusa cover across sites, thus determining whether changes through time were similar among years. One-sample t-tests were used to calculate differences of P. superfusa cover among sites in 2009 and to determine differences between absolute change and proportional change in coral cover between time points. Absolute change was the difference in coral surface area (cm2) per quadrat (e.g., cover in 2010—cover in 2009, or C2010-C2009). Proportional change was the relative difference in coral cover between time points compared to the original coral cover in 2009 (e.g., [C2010-C2009]/C2009). Tukey’s post-hoc tests were used to determine potential differences among sites.

Because the same individual colony can appear in multiple years, analyses assessed the possible effect of independence of colonies through time. ANCOVA was used to determine interaction effects of year and site. Because 109 colonies were repeatedly assessed, data were randomly sampled using each colony only once. A binomial logistic regression was used to determine the effect of colony size on survivorship across years. Analyses were performed using R version 3.1.2 (R Development Core Team, 2014).

Results

The 2009 temperature rise impacted approximately 75% of the Porites superfusa population at Palmyra Atoll with bleaching and mortality observed island-wide. The initial size of individual colonies was greater on average than in subsequent years. The number of colonies present in 2009 was also greater than in later time points. True recruit numbers in 2010 (after the bleaching event of 2009) were lower than in 2011 and 2012 (Table S1).

All sites surveyed exhibited similar patterns of decline in P. superfusa from 2009 to 2010, followed by fluctuations of growth in 2011 and 2012 (Figs. 4 and 5). From 2009 to 2010, P. superfusa mortality was ubiquitous across sites, and the growth for this period was the lowest observed during the study (Fig. 4). Mortality was divided into two categories: complete and partial mortality. These mechanisms contributed similarly to the reduction in cover observed (Fig. 4, partial mortality: −101.8[56.79] cm2 per quadrat, mean [SE], complete mortality: −105.7 [39.10] cm2 per quadrat, mean [SE]). Recruitment and growth were minimal initially, revealing no appreciable difference among growth mechanisms (Fig. 4, Recruit: 4.99 [1.31] cm2, Growth: 4.56 [0.69] cm2 per quadrat). Across the study duration, the highest rates of growth were observed from 2010 to 2011. Resurrected recruits (i.e., apparent recruits that appear in a location where a colony had been recorded as suffering mortality in previous time points) made up a third of total “recruitment” (12.45 [4.45] cm2). Recruitment contributed less than colonial growth to the overall gain in cover (Fig. 4, Recruit: 24.06 [8.77] cm2, Growth: 39.84[13.19] cm2). From 2011 to 2012, resurrected recruits played a decreased role (2.702 [0.49] cm2) in population growth. True recruitment and growth contributed similarly to overall growth (Recruit: 14.29 [18.37] cm2, Growth: 18.37 [6.45] cm2).

Figure 4 Patterns of gain and loss for Porites superfusa during- and post-bleaching event.

(A) 2009 to 2010 change in cover, (B) 2010 to 2011 change in cover, (C) 2011 to 2012 change in cover. ‘Resurrected’ recruits do no exist within the first time transition, as no data exists prior to 2009.

Figure 5 Porites superfusa cover from 2009 to 2012, averaged by site through time, at yearly surveys.

ENSO event occurs in 2009. Sites (denoted by different lines) follow similar patterns through time. Cover decline is followed by variations in recovery and further decline. Figure 2 shows site locations around island.

In 2011, 83 of over 400 colonies (21%) reappeared (after being overgrown by turf and crustose coralline algae in 2010) and began to re-colonize the substrate 10 months later (resurrected recruits, Table S1). Out of 1,150 colonies tracked, approximately 100 fragmented into apparent daughter clones (due to partial mortality) and 79 fused. Of the 100 colonies that fragmented, 52 fused back with a remnant section of the original colony within two years.

This study found that P. superfusa colonies gained upwards of 80 cm2 of new tissue area per year and lost up to 170 cm2 of tissue area per year, with a mean tissue change of —4.75[0.232] cm2— per colony. During the time period that included the bleaching event from 2009 to 2010, P. superfusa declined an average of 8.3 cm2 per quadrat (total quadrat size, 540 cm2). From 2010 to 2011, the corals increased an average of 1.7 cm2 per quadrat (Welch two sample t-test, t =  − 3.74, p = 0.028) and from 2011 to 2012 corals declined an average of 0.9 cm2.

Average change in area, when normalized to initial coral size in 2009, was largely negative (Figs. 4 and 5) and inversely related to size. Smaller corals gained more proportional area relative to larger colonies. The year with the most negative growth was 2009–2010, with the following years slightly less negative (Fig. 5). The overall change in P. superfusa cover from 2009 to 2010 (C2010-C2009) was negative (one sample t-test on C2010-C2009, p < 0.001) with no clear differences among sites (Tukey post-hoc). The change in cover from 2010 to 2011 (C2011-C2010) was minimal growth (one sample t-test on C2011-C2010, p < 0.001), followed by a slightly negative change from 2011 to 2012 (one sample t-test, p = 0.09, Fig. 5). There was no statistical artifact associated with including repeatedly measured colonies, i.e., colonies with multiple transition data through time, in the ANCOVA analysis, as the results from bootstrapped subsampled data were comparable to those from the entire dataset (in only one iteration of the random resampling was a significant site effect was noted).

We recorded a significant relationship between initial size of a coral colony in 2009 and survival across time points (Fig. 6). Larger corals had the greatest declines in area, but they were more likely to survive across time points. The size of the colony in 2009 predicted survival in 2012, with larger colonies showing a higher probability of survival (binary logistic regression, p < 0.05, for all time points). Figure 6 depicts the frequency of colony size classes and their survival from 2009 to 2012. The pattern is bimodal, with an increase in frequencies in the smaller corals, followed by a sharp decrease and then a slight increasing pattern of size. The size frequency between the two groups (survivors and non-survivors) was similar; however, the largest size classes all survived to the end of the study, while the smaller size classes were more abundant across all years (Fig. 6). The average size of colonies varied from approximately 5 cm2 to 14 cm2.

Figure 6 Paired histogram of initial colony size (in 2009) and survivorship (to 2012).

Larger coral colonies were more likely to survive from 2009 to 2012. White denotes coral colonies that died before the 2012 survey. Black denotes coral colonies that survived to 2012.

Discussion

The goal of this study was to quantify the population dynamics of a common encrusting coral during and after an ENSO event on a remote reef in the central Pacific. The effects of the ENSO warm-water event on the population of Porites superfusa on the forereef on Palmyra Atoll were dramatic and widespread. The ENSO event was associated with temperatures up to 30 °C at the sites surveyed in 2009 (Maximum degree heating weeks 16, Williams et al., 2010), and in our results P. superfusa suffered high mortality rates for over a year after the bleaching event.

The fates of colonies tracked through time revealed that the mechanisms of growth and death changed in surprising ways. After the bleaching event in 2009, corals suffered widespread mortality. Although, clonal growth (colony expansion) was recorded in a subset of individuals, complete and partial mortality were sufficiently large so as to overcome the signal of minimal growth (Table S1). Regrowth (from colony expansion and resurrected recruits) was an important driver of recovery and contributed to over 50% of the increase in cover of P. superfusa. Growth of corals in cryptic habitats, such as crevices or areas shaded by other corals, may be responsible for some of this survival and growth. In 2010, after the mortality event, colony growth increased. This is consistent with other documented case studies, which suggest that corals may increase growth rates to heal after injury (e.g., Kramarsky-Winter & Loya, 2000. However, the recorded increase in growth contradicts the idea that bleaching may decrease regeneration abilities of some corals (Meesters & Bak, 1993).

Resurrected recruits played a surprising role in the coral’s population growth (over 12% of total growth). The resurrected recruits could be a type of regrowth of cryptic tissue, beyond the observational scope of this study. Some of the regrowth appeared to be emerging out from underneath coral species that were shading the underlying benthos, such as Pocillopora meandrina, and this survival could indicate that partial mortality was induced from UV stress in exposed portions of the benthos (Baird & Marshall, 2002). However, in many cases, the resurrected recruits appeared on a flat reef surface, emerging from benthic areas apparently covered in turf or crustose coralline algae. Unfortunately, the resolution of the photographs did not allow us to determine without question how the corals may have resurrected, for example, from a surviving piece of tissue in a small crevice or otherwise obscured from view. Alternatively, it is possible that these patterns represent true recruitment of larvae onto the exact same location as an adult colony had formerly occupied. Nonetheless, these observations hint at the modularity of coral colonies, and the spatial consistency among ‘new’ recruits is intriguing and warrants further investigation.

In certain cases the partial mortality of P. superfusa resulted in fragmentation (100 colonies) or clone fission (79 colonies). Fragmented coral colonies living in close proximity (groups of daughter clones) have a potentially higher rate of survival, as they are not as easily eliminated from disease or competition (Highsmith, 1982). However, fragmentation in this study was relatively low (<10%), and thus it is unclear if the “fused” colonies (<10%) were more likely to survive. While recruitment is important for some coral recovery, within-colony expansion and regrowth were of comparable quantitative importance regarding P. superfusa growth at Palmyra Atoll. Some Porites spp. have been found to have a limited capacity for recruitment (Potts et al., 1985), which may make regrowth a specifically useful recovery mechanism for this genus. A recent study by Roff and colleagues (2014) found that regrowth was an important contributor to the population-scale recovery of massive Porites. The ability of corals to regrow from remnant polyps may prove vital to recovery as climate change continues. If the regrowth of smalll fragments of remnant tissue can bring colonies back from apparent mortality, then colonies that appear dead in traditional coral surveys may actually have a chance at survival. As a result, traditional coral surveys may overemphasize mortality.

Coral colony size was an important predictor of the coral’s ultimate fate (Fig. 6). Interestingly, P. superfusa seemed to have size-dependent growth and death. Growth decreased with increasing size, suggesting that this species may have determinate growth. This growth pattern could be due to high levels of partial mortality. In addition, as is consistent with the literature, smaller colonies of P. superfusa experienced higher rates of overall change, including mortality and recovery (Hughes & Jackson, 1985). Large colonies are more likely to suffer partial mortality, which may be part of the declining growth with size phenomenon (Hughes & Jackson, 1985). Probabilistically, larger colonies are more likely to suffer injury due to a larger surface area, and this may be a factor in the decline of overall growth. Smaller colonies tend to experience damage in a more binary way: either resisting disturbance entirely or dying completely, suffering less incidence of partial mortality (Connell, 1973; Hughes & Jackson, 1985).

While this study documents the potential for coral regrowth following anomalous temperature events, it is important to note that recovery did not exceed mortality over the course of this study. The decline in P. superfusa cover from 2009 to 2010 occurred more rapidly than the increase in coral cover from 2010 to 2011 and the decline from 2011 to 2012. Full population recovery is a slow process, and the longer-term trajectory of P. superfusa on Palmyra remains to be seen.

Understanding the effects of large-scale phenomena on community dynamics in relatively pristine reefs provides critical benchmarks for coral demography (Edmunds, 2002). Further studies on this topic will help quantify the importance of fragmented or remnant corals for reef recovery processes. Examination of other species at these sites may provide important comparisons among morphologies and between species with different life history strategies, which combined may help infer the likelihood of community level recovery. The causes of bleaching, mortality, resistance, and recovery are clearly complex and species-specific. Thus, additional studies from remote Pacific island coral communities are important for understanding the capacity of these systems for recovery via different mechanisms; such studies will help to directly improve the policies for protection of reefs worldwide.

Conclusions

This study documented a bleaching event and subsequent change in areal cover of a common coral on a remote central Pacific atoll. Coral populations can change via four different dynamic processes: complete mortality, partial mortality, growth, and true larval recruitment. This study suggests a fifth dynamic, ‘resurrected’ recruits, as a mechanism contributing to coral recovery. Additional research is needed to determine the source of this coral growth (e.g., exploitation of cryptic habitats, regrowth of remnant tissue).

With increasing climate pressures, many coral reefs have experienced rapid decline. In such cases, tissue loss due to mortality is faster and more obvious than tissue gain due to growth. Coral growth is often slow and subtle, making it a difficult research and management target for short time scales. Despite the challenges associated with quantification, regrowth is a critical process in coral recovery, and it is vital we understand the mechanisms controlling it. The implications from this study of colony-specific patterns of decline and recovery should be scaled up to examine the role of regrowth and the future of remote Pacific islands’ reef recovery.

Supplemental Information

Table S1 Table of Porites superfusa demographic changes tallied by year and number of colonies

Year transitions (2009–2010, 2010–2011, 2011–2012) are recorded because changes in numbers of colonies occurred from one year survey to the next. No resurrected recruits were counted in the 2009 to 2010 year transition because in order to ‘resurrect’ the colony must have at least three years of study. All rows are in number of colonies, except for the mean initial size of colonies in cm2.

Click here for additional data file.

Supplemental Information 2 Raw data of coral colony sizes over time, at each site

Click here for additional data file.

Supplemental Information 3 All statistical analysis code completed in R

Click here for additional data file.

Supplemental Information 4 Code for Tukey’s post hoc

Click here for additional data file.

Special thanks to Ben Knowles, Genivaldo Gueiros, Clint Edwards for their edits on the manuscript. Field work completed with the help of Maggie Johnson and Amanda Carter.

Additional Information and Declarations

Competing Interests

Author Contributions

Field Study Permissions

Data Availability

The authors declare there are no competing interests.

Kathryn Anne Furby conceived and designed the experiments, performed the experiments, analyzed the data, wrote the paper, prepared figures and/or tables, reviewed drafts of the paper.

Jennifer Ellen Smith and Stuart Adrian Sandin conceived and designed the experiments, analyzed the data, contributed reagents/materials/analysis tools, reviewed drafts of the paper.

The following information was supplied relating to field study approvals (i.e., approving body and any reference numbers):

United States Fish and Wildlife Special Use Permit #12533-16006.

The following information was supplied regarding data availability:

The raw data has been supplied as a Supplementary File.

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
