# Peer review of "Porites superfusa mortality and recovery from a bleaching event at Palmyra Atoll, USA"

_PeerJ, doi:10.7717/peerj.3204_

## Round 0.1 · original submission · Major Revisions

I have heard back from two reviewers, both of whom have offered detailed and constructive comments that will help you improve your paper; please read these over carefully. Therefore, my decision is that "major revisions" are needed.

Reviewer 1 ·

Basic reporting

The submission adheres to PeerJ policies and is written in good English. Research is well reported, with a good introduction, clear material and methods, well-presented results and and a not excessive discussion (which could be significantly strengthened, however). The figures are good and needed, but Fig. 5 (a histogram) is not what is promised in the text (a logistic regression).
Reporting is acceptable for publication and the type of article.

Experimental design

The research design is well described and the methods are clear. It would be possible to repeat the experiment, if needed. There are no major concerns with regards to the approach to data gathering and/or data analysis. Data gathering appears to have been as rigorous as possible, and data evaluation is quite rigorous. More could have been made of the data, but it is what it is.

Validity of the findings

Findings are valid. Data are robust, statistically sound and controlled. Conclusions are well stated. Many more results could have been produced and much more advanced analyses have been performed (even the standard textbooks go much further than what was done here) with data of this quality, but so be it.

Additional comments

Intro, line 41, 42: the sentence isn't logical
line 48: delete "demographic". Throughout the paper much is made of demography and the word is dropped everywhere. Only, I see no demography - which is a very precise and quantitative subject - being done. There is some basic work-up of data, but that's really it.
Material and methods:
line 63-4: How was this species an obvious loser? This should be strengthened with some data or at least something beyond a mere statement.
line 71: delete "on the benthos"
line 87: change to Abramoff et al, 2004
line 107: delete "if" (4th word)
Results
line 146: introduce "of" after "Out"
line 163. introduce a period after ANCOVA
line 168. There is something wrong here. Fig. 5 is a histogram, not a logistic regression as promised here. I would surmise that a graph is missing.
Discussion
line 173-4: I would delete the discussion's second sentence. It is a gross overstatement and demonstrates only, if anything, that the authors have done remarkably little reading of the literature and thus missed a lot of key references. And they would have found that this paper is not so unique...
line 178: the sentence is unclear. What is "genera prevalence" supposed to be. Where and how do they prevail?
line 204: introduce a "may" here. What is dscribed here is not a given.
line 218/9: "growth rates decelerated"...really? Well, linear extension rates in Porites are constant, there are dozens of papers that show that (why else would this genus be the favorite of all these papers looking for a deceleration due to OA etc?). There may be an apparent deceleration of percent areal increase when looking straight down on the coral...but that is a misrepresentation due to the "flat" representation of the coral. Doing just a little math will show that, even in a flat colony with an area of pi*R^2, the increase in area MUST be exponential. If that is not so, it is due to exponentially increases partial mortality...and interesting finding but not what is reported here. If such a mechanism is at work, this needs to be better developed and shown - else, this seems like a somewhat unreflected statement.
line 223: not only statistically, but ALWAYS, do larger corals have larger surface areas. That's because they are larger...duh. Or did I miss something? If so, please explain better.
line 224: As far as I know, there is no such thing as "binary demographics". I think a better word could be found...please make an effort to do so.
line 234/5: this is not only true for damaged reefs, but for healthy ones as well.
line 241: what is "policy protection"?
Conclusion
Delete in its entirety and rewrite. The conclusion is not meant to be a "stream of consciousness" but a concise compilation of key points of the study.

Fig. 5: despite the fact that what is shown is not the promised logistic regression is interesting and could be used for at least some analysis. The bimodality clearly indicates overlap of two (clearly not normally distributed - which is interesting in and of itself) generations. This is unusual in corals and really should be explored - why could that pattern arise? But just like this example, many other potentially worthwhile and fruitful avenues of examination are being ignored, which makes this paper a rather basic and unexciting read.

Reviewer 2 ·

Basic reporting

This study investigates the effects of the 2009 coral bleaching event on the demography of the coral Porites superfusa at Palmyra Atoll over a three-year period (2009-2012). While the manuscript is interesting and relevant, the terminology and language used throughout the manuscript is ambiguous, making it difficult to follow and to assess whether the results are robust. This mainly stems from inconsistent use of demography categories throughout the manuscript (see examples in the ‘general comments’ section below) and could be addressed by clearly naming categories and consistently using these category names throughout the entire manuscript. For ease of understanding and consistency with the coral demography literature, I recommend renaming the category ‘loss’ to ‘partial mortality’. While the introduction section is well written and clearly sets up the study, the discussion section does not sufficiently place study results in the context of the wider demography literature and findings for other species and/or other locations.

Experimental design

More clearly explain why you chose the 4 study sites and at what point during the bleaching event the initial surveys were conducted in 2009 (i.e. at the beginning, peak etc. of bleaching). Also, the manuscript title refers to mortality and recovery from bleaching, yet the extent/degree of bleaching of Porites superfusa colonies recorded in quadrats was not assessed (or at least results are not given in the manuscript). I urge the authors to include analyses that assess recovery trajectories of coral colonies as a function of their bleaching extent in a revised submission.

Validity of the findings

Due to the inconsistent use of demographic categories the robustness of research findings is difficult to assess (see above). I recommend including a table that gives the number of colonies in quadrats for each of study year, the average colony area, etc., which, along with more consistent terminology, should make it easier to follow. The use of subheadings and more systematic structuring of results (e.g. complete mortality, partial mortality, growth, recruitment) could also aid understanding.

Additional comments

Line 37: List the few studies that you refer to

Line 51: Typo in Porites astreoides

Lines 53-58: I would lead with the broad goal, followed by the specific aims.

Line 63: Explain what you mean by weedy coral – are you referring to the Darling et al. (2012) life-history-strategies?

Line 64: Define / quantify what you mean by ‘obvious loser’. See comment above re quantifying bleaching extent of coral colonies in 2009.

Lines 89 ff: Change to ‘complete mortality’ and ‘partial mortality’ and use the 5 category names consistently throughout. You already use the term ‘partial mortality’ in the abstract and discussion.

Line 94: Specify that growth was measured in cm2.

Line 107: Delete the word ‘if’

Line 109: When you refer to ‘growth rate’ later on, do you mean the ‘proportional change’ that you describe here?

Line 114: ‘Some colonies’ is vague. Be specific and quantify nr or percent of colonies.

Line 116: Report the ‘family’ (binomial?) you used in your logistic regressions. Also specify which is the response or reword to ‘to determine the effect of colony size on survivorship’ to make clear that survivorship is the response.

Lines 122-125: Move to the introduction

Lines 125-126: ‘Noticeable impact’ is vague. Be specific about bleaching extent.

Line 128: Move to the methods section

Lines 129 ff: I would start the results by giving details of how many colonies you recorded in surveys each year etc. The overview table I suggested above.

Line 132: It would make more sense to say ‘Mortality was divided into two categories: complete and partial mortality. Best to also specify this in the methods.

Line 134: Specify whether the numbers in [ ] refer to standard errors or standard deviation?

Lines 137-142: Inconsistent use of ‘recruitment’. When you say ‘total recruitment’, do you mean ‘resurrected recruitment’ + ‘true recruitment’?

Line 144: Say ‘84 out of 400 colonies (21%) reappeared etc’. Also, does this mean that these 84 were ‘resurrected recruits’? If so, then specify this in brackets. Are ‘daughter clones’ are the result of partial mortality?

Line 150: Is mean tissue change of 4.75 cm2 absolute change, as I believe that overall you recorded a decline, right?

Line 155ff: Fig 4 only shows year on year change and not change over the entire period (2009-2012); I suggest adding overall change for the study period to this Figure or otherwise in a separate table or Figure.

Lines 163-164: Quantify what you mean by ‘some iterations’ and ‘was very rare’.

Line 165: Better to say: ‘We recorded a significant relationship between initial colony size in 2009 and survival’

Lines 166-167: ‘more negative growth rates’ … do you mean greatest areal declines? The sentence ‘Size of the colony in 2009 etc’ is a repeat of what you said in line 165.

Line 178: I would follow on after the Williams reference with ‘and our results show that P. superfusa suffered mortality …’

Line 181: ‘just after the bleaching event in 2009’ … do you mean the 2009 or 2010 data point?

Lines 183 to 189: Clarify what you mean by ‘regrowth’ here; ‘growth’ or ‘resurrected recruits’?

Line 184: Give examples for cryptic habitats

Lines 189-191: This is unclear as you refer to ‘regrowth’ at the start of the sentence and then to ‘partial mortality’ towards the end

Line 195: How can anything grow underneath a massive colony, as their base in encrusting?

Lines 203-207: Quantify ‘some cases’. I also think that ‘fused’ is the wrong term here. Do you mean colonies that have come about through partial mortality?

Line 208: Unclear use of terms ‘growth/regrowth’

Line 215: Can you expand on what these ‘interesting implications’ are?

Line 220: How does this compare with other studies for other species or for Porites in other locations?

Line 228: Apologies if I missed this, but you aren’t showing them anywhere in the manuscript. Please present these results.

Lines 239-241: Either expand on this or delete this sentence, which seems more of an afterthought.

Line 246: Your study currently does NOT document the bleaching and I suggest that you present bleaching extent of colonies in the results section.

---

## Round 0.2 · Major Revisions

I have gone over your revised submission myself, and the manuscript is much improved. Still, echoing the first round of reviewers' comments, there are some areas in which the wording is inexact or imprecise. Additionally, I have added two or three comments that need addressing, in particular regarding the "resurrected recruits" and a few of the Figures. Thus, my decision is "major revisions" are required, although not nearly on the scale of the previous version. You can consider this decision to be somewhere in between "major" and "minor". For details, please see the attached PDF, and if you have any questions, feel free to contact me via PeerJ or directly (e-mail). I look forward to seeing a revised version.

---

## Round 0.3 · accepted · Accept

The manuscript has been thoroughly and well edited - and is now acceptable for publication. I look forward to seeing the published version.